# Microsatellite Instability in Russian Patients with Colorectal Cancer

**DOI:** 10.3390/ijms23137062

**Published:** 2022-06-25

**Authors:** Vitaly Shubin, Yury Shelygin, Sergey Achkasov, Oleg Sushkov, Ilya Nazarov, Alexey Ponomarenko, Iuliia Alimova, Anna Loginova, Aleksey Tsukanov

**Affiliations:** Ryzhikh National Medical Research Center of Coloproctology, 123423 Moscow, Russia; shelygin_ya@gnck.ru (Y.S.); achkasov_si@gnck.ru (S.A.); sushkov_oi@gnck.ru (O.S.); nazarov_iv@gnck.ru (I.N.); ponomarenko_aa@gnck.ru (A.P.); alimova_yv@gnck.ru (I.A.); loginova_an@gnck.ru (A.L.)

**Keywords:** microsatellite instability, colorectal cancer, Lynch syndrome

## Abstract

The aim of this study was to determine the characteristics of Russian patients with microsatellite instability (MSI) tumors. MSI in the tumor was determined in 514 patients with colon cancer using PCR and subsequent fragment analysis for five markers (NR21, NR24, BAT25, BAT26, and NR27). In the presence of microsatellite instability, the mismatch repair (MMR) system genes were examined using the NGS and MLPA methods to establish the diagnosis of Lynch syndrome. The overall frequency of MSI tumors was 15%: at stage I—19% (9/48), at stage II—21% (44/213), at stage III—16% (26/160), and at stage IV—2% (2/93). Patients with MSI tumors differed in the age of diagnosis, tumor localization, time of cancer recurrence, and stage of the disease. The overall and disease-free survival of patients whose tumors had MSI status was higher than that of patients with microsatellite-stable status, *p* = 0.04 and *p* = 0.02, respectively. Analysis of overall and disease-free survival of patients with Lynch syndrome and patients with sporadic colon cancer, but with MSI status, did not reveal significant differences, *p* = 0.52 and *p* = 0.24, respectively. The age of patients with Lynch syndrome was significantly younger than that of patients with sporadic colon cancer whose tumors had MSI status (*p* < 0.001).

## 1. Introduction

In 1993, it was found that some colon tumors are characterized by genome instability—namely, mutations occur in repetitive DNA regions, usually deletions/insertions of one or more nucleotides [1,2]. Short (up to six nucleotides), repeated DNA fragments are usually called microsatellites, and the phenomenon arising as a result of disruption of the unpaired DNA base repair system is called microsatellite instability (MSI) [2]. According to the 2019 ESMO guidelines, a panel consisting of two mononucleotides (BAT-25 and BAT-26) and three dinucleotide markers (D5S346, D2S123, and D17S255), or an alternative panel including five mononucleotide markers (BAT-25, BAT-26, NR-21, NR-24, and NR-27), should be used to determine MSI status in patients with colorectal cancer (CRC) [3]. Goel A. showed that it is not necessary to use a control sample of normal tissue or blood to diagnose MSI using PCR based on five mononucleotide markers [4]. It was recommended to abandon the concepts of high- and low-level microsatellite instability—MSI-H and MSI-L, respectively. In this case, tumors with MSI-L should be equated to MSS, and MSI-H should be designated as MSI [3]. Determination of MSI status in a tumor allows assessing the feasibility of prescribing adjuvant chemotherapy in stage II CRC, provides an opportunity to verify the category of patients for whom immunotherapy can be effective, and also allows one to suspect Lynch (LS) and constitutional mismatch repair deficit (CMMRD) syndromes [5,6,7]. In addition, Fujiyoshi K. showed that the MSI status has an important prognostic value at stage IV of CRC—namely, the detection of MSI in the tumor of patients with hematogenous metastases is associated with a low survival rate, while in patients with peritoneal carcinomatosis (PC), conversely, the survival rate is higher, 1.33 vs. 5.2 years (*p* = 0.006) [8].

However, it should be borne in mind that population studies demonstrate different frequencies of MSI (from 8.8% to 20.3%) and phenotypic manifestations among patients with CRC [9,10,11,12,13,14]. It is also important to note that the frequency of MSI can also differ among the stages of colorectal cancer, and its presence may help in the treatment of this disease.

The aim of the study was to determine microsatellite instability among Russian patients with stage I-IV colorectal cancer.

## 2. Results

The frequency of MSI in the studied sample of patients with CRC was as follows: at stage I—19% (9/48), at stage II—21% (44/213), at stage III—16% (26/160), and at stage IV—2% (2/93). Accordingly, the MSI frequency is revealed by the formula F(MSI) = ((I% + II% + III% + IV%)/4), where F(MSI) is the calculated MSI frequency, and I%–IV% are the MSI frequencies at the corresponding stages of CRC; it was found to be 15%.

### 2.1. Clinicopathological Features According to MSI Status

The differences in age, location, time of tumor detection, and CRC stage in patients with the MSI and MSS status were statistically significant (Table 1).

### 2.2. MSI Status and Patients Survival

The median follow-up was 27 (2–126) months. An analysis of overall survival (OS), conducted on 469 patients, showed that 61 (13%) CRC patients with MSI,] had better OS than patients with MSS status (*p* = 0.04). Thus, the 1-, 3-, and 5-year survival rates of patients with MSI tumors were 96%, 86%, and 85%, and those with MSS status were 95%, 85%, and 64%, respectively (Figure 1).

Of the 468 patients included in the survival analysis, 91 (19%) patients had a recurrence. MSI status showed as a good prognostic factor for disease-free survival. Therefore, 1-, 3-, and 5-year DFS in patients with MSI tumor status was 96%, 89%, and 53%, and with MSS status, these values were 96%, 81%, and 51%, respectively (*p* = 0.02) (Figure 2).

### 2.3. Comparison of Lynch Syndrome and Non-Lynch Syndrome with MSI

As a result of the comparative analysis, it was revealed that patients with Lynch syndrome were significantly younger than MSI sporadic patients: 43 ± 13 vs. 53 ± 14 (*p* < 0.001) (Table 2).

### 2.4. Survival in Lynch Syndrome/Non-Lynch Patients

The overall and disease-free survival rates of patients with Lynch syndrome and patients with MSI sporadic tumor status did not differ, *p* = 0.52 and *p* = 0.24, respectively. (Figure 3A,B).

### 2.5. Analysis of Risk Factors of Patients with MSI Status

Univariate and multivariate analyses showed that the risk factors for the presence of MSI in a tumor in patients with CRC were young age and tumor localization in the right colon. It was also shown that the development of metachronous cancer was a risk factor for patients with MSI (Table 3).

## 3. Discussion

At the end of the last century, it was established that at least 15% of patients with CRC have MSI tumors. Most often, MSI tumors are localized in the right parts of the colon and relatively rarely metastasize [15]. However, a number of studies show that the incidence of MSI in tumors in colorectal cancer varies from 8.8% to 20.3% [9,10,11,12,13,14]. The frequency of microsatellite instability revealed by us, 15%, is comparable to China (14%) [12], France (14%) [11], and the USA (15%) [13]. As our sample could be biased, we conducted a comparative analysis of the frequency of our sample with the frequency of patients with colorectal cancer of different stages in the population and did not find statistically significant differences (*p* > 0.05). In Russia, according to Kaprin et al., among all patients with colorectal cancer, stages I–II accounted for ~51% of cases, III—23%, and IV—24% [16]. Our sample fits this distribution, so the resulting MSI frequency reflects population characteristics. It should be noted that the frequency of MSI obtained by us in tumors of patients with CRC at stages I–III varied from 16% to 21%, and at stage IV, it plummeted to 2% (Figure 4). Although in a number of countries, researchers have shown that the frequency of MSI does not differ depending on the stage (*p* = 0.06) [9], and in a number of other countries, statistically significant differences have been found (*p* < 0.001) [10].

We performed a cross-population analysis of the incidence of MSI for different stages of colon cancer. It was found that in stage II, the frequency of MSI (21%) was lower than that in the German population (30%) (*p* = 0.02), but in stage III, conversely, it was higher, 16% vs. 9% (*p* = 0.03). At the same time, the frequency of MSI at stage IV CRC (2%) found by us in the Russian population was lower than that in the German (7%) and significantly lower than that in the Italian (17%) (*p* < 0.01) populations. What is the reason for such a significant difference? It is difficult to say at present, and this issue requires further investigation.

According to the literature, the age of CRC patients with MSS and MSI status, as a rule, does not differ [9,12,14]. In our case, the age of CRC patients with MSI (49 ± 15 years) was significantly lower than that of patients with CRC with MSS status (56 ± 13 years) (*p* < 0.001), in connection with which we analyzed the group of patients with MSI status.

In our opinion, statistically significant differences in age are associated with the fact that our analysis included patients with Lynch syndrome. Our assumptions were confirmed when patients with Lynch syndrome were excluded from the comparison. Thus, we found no differences between the age of patients with sporadic cancer, but with MSI status, which was 53 ± 14 years, and the age of patients with microsatellite stable tumors (Figure 5).

The authors from different countries have established the fact that MSI is more common in patients with tumor localization in the right half of the colon [10,14], which is also consistent with our results. However, the frequency of MSI obtained by us with the localization of the tumor in the left colon was higher than, for example, in Germany and Italy, *p* < 0.001 and *p* = 0.02, respectively (Figure 6). It can be assumed that the higher frequency of MSI in neoplasms located in the left regions of the colon may be associated with the characteristics of the Russian population.

Most of the studies examining MSI with CRC do not estimate such a factor as the time (age of the patient) of its detection. At the same time, Velayos et al. showed that the presence of MSI in metachronous tumors is due to hereditary predisposition, and synchronous tumors are sporadic [17]. According to our data, despite the fact that metachronous tumors are more common in patients with MSI, they were found with approximately the same frequency in both those with Lynch syndrome and patients with sporadic cancer (*p* = 0.47). Therefore, this fact most likely indicates the absence of the influence of heredity in Russian patients with microsatellite instability CRC on metachronism.

Patient survival is one of the important indicators. In a systematic review, Toh showed that MSI is a factor of good prognosis in stages I–III of disease while emphasizing the fact that neither chemotherapy nor immunotherapy affects survival rates in such patients [18]. However, in patients in stage IV with CRC and with MSI tumors, survival rates are higher if the patients were treated with immunotherapy [3]. In our study, we also showed that patients with MSI status had better OS and DFS (Figure 2 and Figure 3). Focusing on Toh’s study, we performed a survival analysis excluding patients with stage IV CRC. It was revealed that the survival rate of patients with MSI and MSS status in stages I–III did not differ (*p* = 0.53) (Figure 7). It can be assumed, that this may be due to the short duration of observation of our patients.

## 4. Materials and Methods

### 4.1. Patients

In the period from January 2016 to December 2019, the Department of Laboratory Genetics examined 514 tumor tissue samples from patients who received surgery for CRC for the presence of microsatellite instability. All patients were older than 18 years. History of chemotherapy and/or radiation therapy, detection of more than 20 polyps in a patient with a malignant tumor [19], and the presence of inflammatory bowel disease served as criteria for non-inclusion of the patient in the study. Tumors were studied for the presence of microsatellite instability in the patients for several reasons—namely, to exclude the appointment of chemotherapy at an early stage and to determine the possibility of prescribing immunotherapy if a patient is suspected of having Lynch syndrome, depending on the patient’s personal desire. Patients did not receive neoadjuvant treatment. Patients received adjuvant therapy according to the Russian practice guidelines for the drug treatment of colon cancer and rectosigmoid junction and rectal cancer published in “Malignant tumors Russian Society of Clinical Oncology” [20]. It was possible to follow up with 469 out of 514 patients (Table 4). The median follow-up was 27 (2–126) months. Informed consent was obtained from all patients included in this study. Our study involved human specimens or tissue. All patients were followed up according to the trial protocols and provided written informed consent. This study conformed with the ethical principles of the Declaration of Helsinki and was approved by the ethical committee of the Ryzhikh National Medical Research Center of Coloproctology, Russian Federation (application number 33A, 14 December 2015).

### 4.2. DNA Extractions

Morphologically verified tumor tissue samples fixed in paraffin blocks were dewaxed with xylene. DNA was then isolated using the QIAamp DNA FFPE Tissue Kit (QIAGEN, Hilden, Germany) according to the manufacturer’s protocol. DNA with a concentration of at least 2 ng/μL eluted in 100 μL of ATE buffer was used in this study. DNA from 200 μL of blood was isolated using a MagNaPure Compact Automatic Nucleic Acid Isolation Station (Roche, Basel, Switzerland) and a MagNa Pure Compact Nucleic Acid Isolation Kit I (Roche, Switzerland). The amount of DNA was measured with a DeNovix QFX (Denovix, WILM, USA) instrument using a Qubit dsDNA BR Assay Kit (ThermoFisher, Waltham, MA, USA).

### 4.3. MSI Analysis

Determination of the status of microsatellite instability was carried out via fragment analysis on an ABI PRISM 3500 device (8 capillaries; 50 cm; Applied Biosystems, Waltham, MA, USA) using five mononucleotide markers (NR21, NR24, NR27, BAT25, and BAT26) (Table 5). DNA fragments were amplified using polymerase chain reaction (PCR). The reaction mixture for PCR (25 μL) included qPCR mix—HS (5x) (Evrogen, Moscow, Russia)—5 μL, H_2_O (free DNA, RNA)—17 μL, primers (F + R) (0.1 μM)—1 + 1 μL (Table 2), and DNA sample (1–10 ng)—1 μL. PCR conditions: 95 °C—5 min; 45 cycles—95 °C—30 s, 56 °C—30 s, 72 °C—10 s; 72 °C—2 min; and 4 °C—storage.

### 4.4. NGS

All patients with colorectal cancer and with MSI status were investigated in terms of mismatch repair (MMR) system and *EPCAM* gene mutations. The search for mutations in the MMR and EPCAM genes was carried out via high-throughput sequencing (NGS) on a NextSeq550 platform (Illumina, San Diego, CA, USA) with enrichment of exome regions, using the TruSeq Exome protocol with IDT xGen Exome v1 probes and subsequent verification via Sanger sequencing on a 3500 ABI PRISM (Applied Biosystems, Waltham, MA, USA) genetic analyzer.

### 4.5. MLPA

Detection of large deletions/duplications in MMR genes was accomplished by using the multiplex ligation-dependent probe amplification (MLPA) method with a set of SALSA MLPA Probemix P003 MLH1/MSH2 reagents (MRC-Holland, Amsterdam, The Netherlands), according to the manufacture’s protocol. The separation of product fragments was carried out using a genetic analyzer, ABI PRISM 3500 (Applied Biosystems, Waltham, MA, USA). The software Coffalyser.Net provided by MRC-Holland (The Netherlands) was used for data analysis.

### 4.6. Statistical Analysis

Statistical analysis was performed using the Statistica 13.0 software. The criterion of maximum likelihood χ2 (M–L chi-square) was used; in the case of comparing two samples, the result was significant at *p* < 0.05, three—*p* < 0.017, and four—*p* < 0.0125. Overall survival of patients was calculated from the date of radical surgery for colorectal cancer to the date of the last observation/death. Disease-free survival was calculated from the date of surgery for colon cancer to the date of recurrence detection. Survival was analyzed according to the Kaplan–Meier method and compared with the log-rank test.

## 5. Conclusions

The study we performed made it possible to establish the characteristics of Russian patients with MSI tumors. The MSI rate was 15%. At the same time, the frequency of MSI in tumors of patients with stages I–III CRC varied from 16% to 21%, and at stage IV, it sharply decreased to 2%. Hereditarily burdened CRC patients with MSI tumors were younger than patients with sporadic microsatellite instability colorectal cancer. In the case of left colon CRC localization, the frequency of MSI was higher in the Russian population than that in other populations. The data obtained can be used for the diagnosis of MSI and the selection of treatment tactics in accordance with the Russian recommendations for the treatment of patients with colorectal cancer.

## Figures and Tables

**Figure 1 ijms-23-07062-f001:**
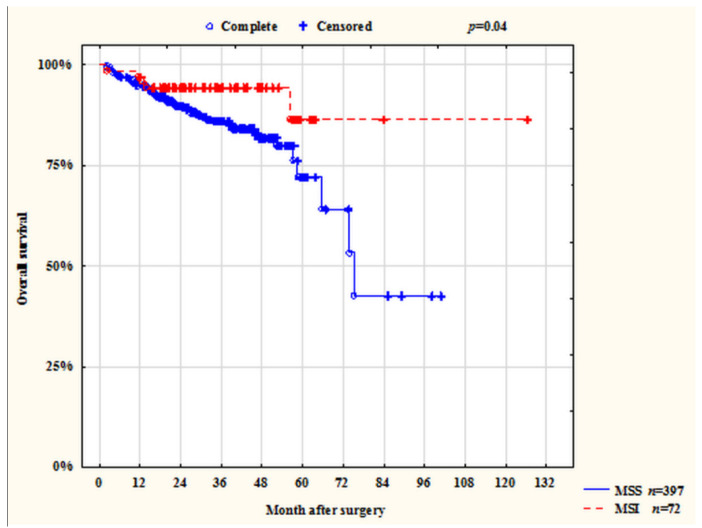
Overall survival of patients with CRC (stages I–IV) depending on the status of microsatellite instability.

**Figure 2 ijms-23-07062-f002:**
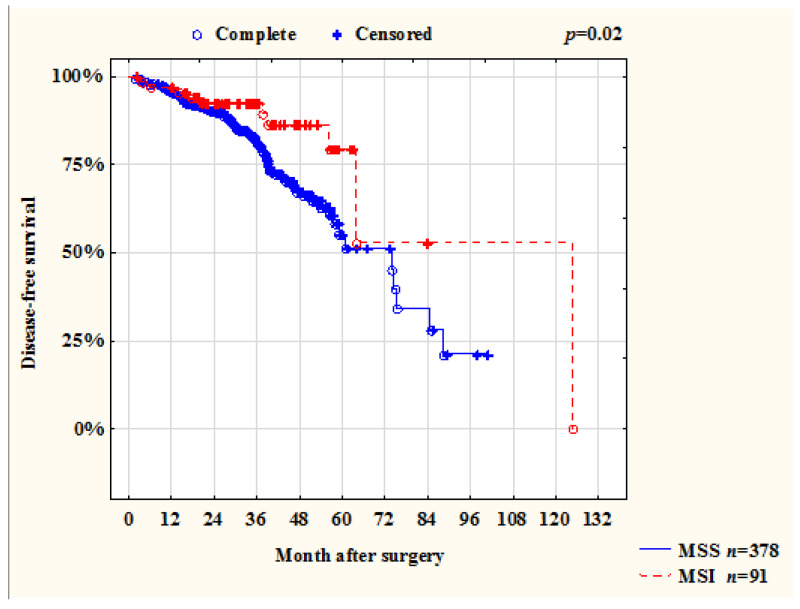
Disease-free survival of patients with CRC (stages I–IV) depending on the status of microsatellite instability.

**Figure 3 ijms-23-07062-f003:**
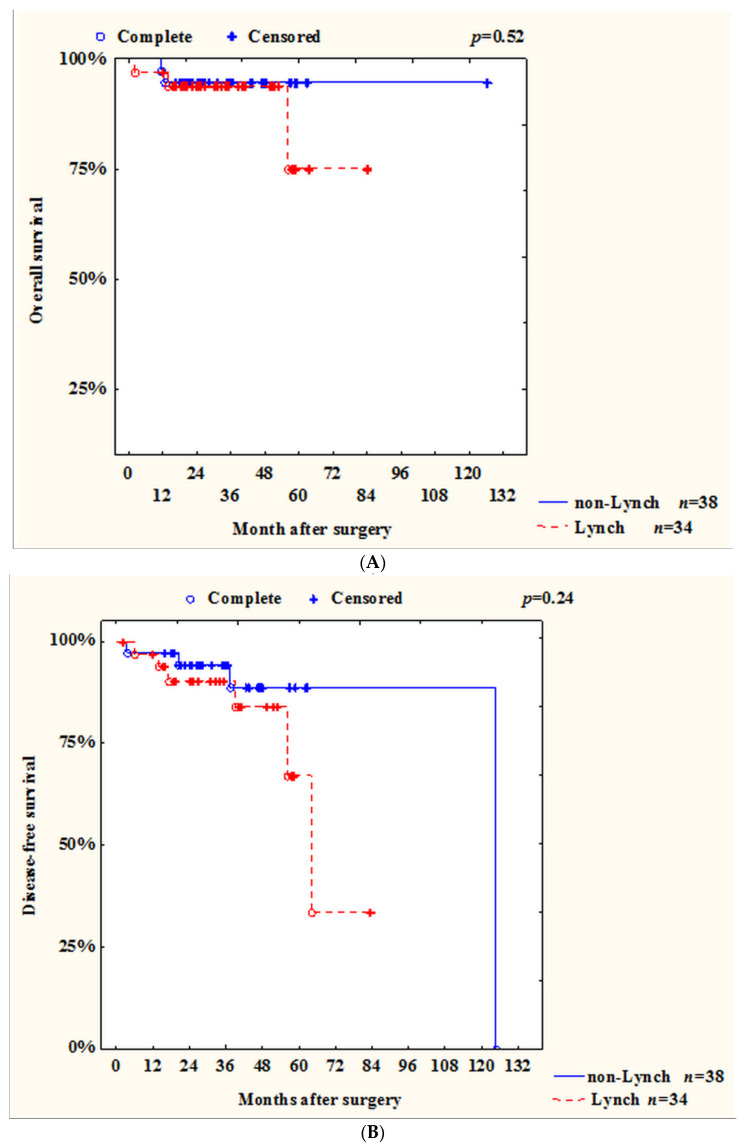
Rates of overall and disease-free survival in patients with Lynch syndrome and patients with MSI status of sporadic tumor: (**A**) overall survival; (**B**) disease-free survival.

**Figure 4 ijms-23-07062-f004:**
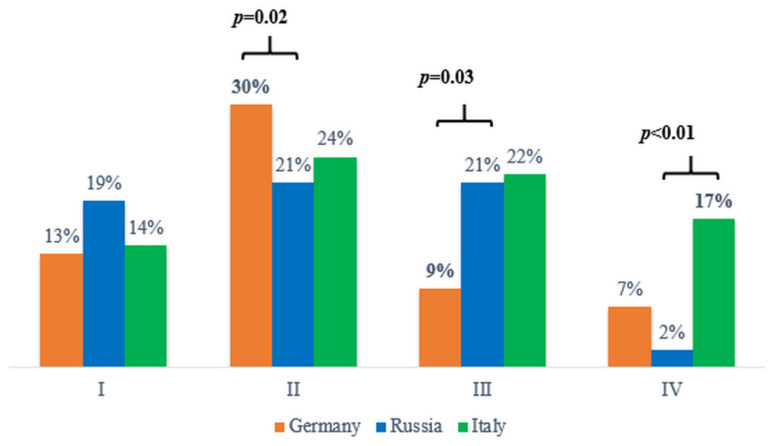
MSI frequencies at stages I–IV CRC.

**Figure 5 ijms-23-07062-f005:**
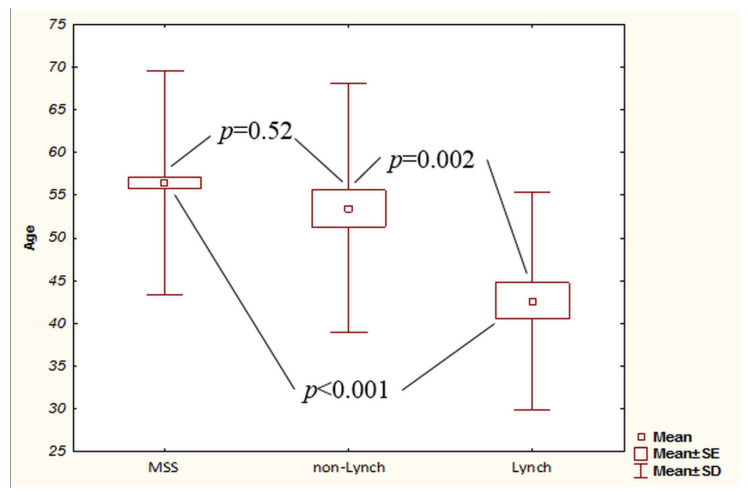
Age of patients whose tumors were MSS, MSI with Lynch’s syndrome, and MSI in sporadic cancer.

**Figure 6 ijms-23-07062-f006:**
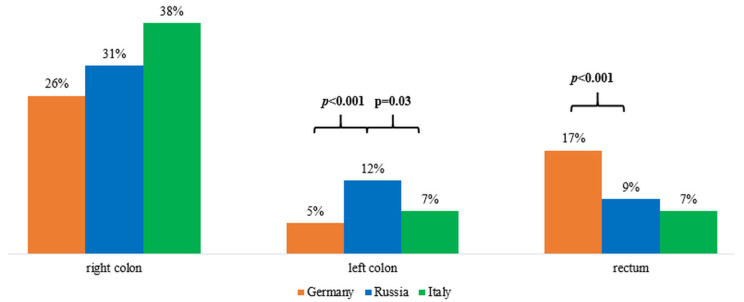
Frequency of MSI in the right colon, left colon, and rectum in Germany, Russia, and Italy.

**Figure 7 ijms-23-07062-f007:**
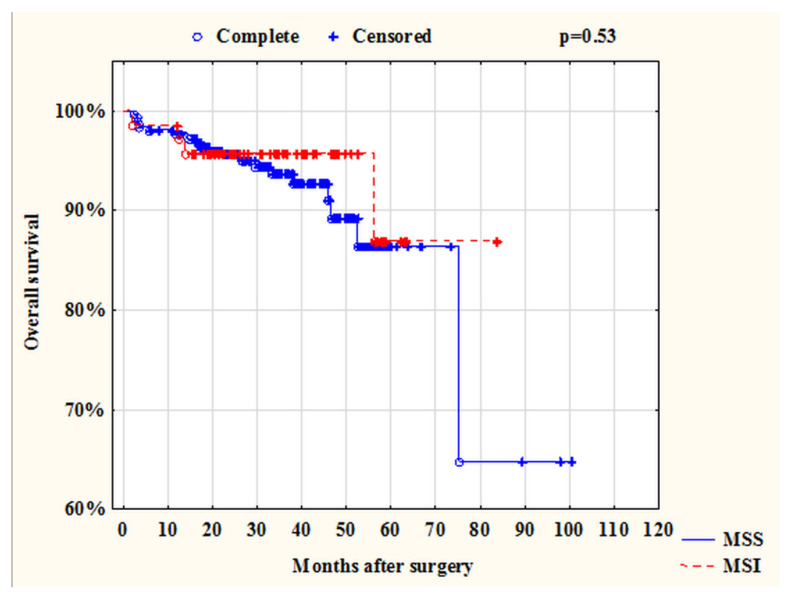
Overall survival of patients with CRC (stages I–III) depending on the status of microsatellite instability.

**Table 1 ijms-23-07062-t001:** Patient’s characteristics according to MSI/MSS status.

	Total *n* = 514	MSS *n* = 433	MSI *n* = 81	*p*-Value
**Age, mean ± SD, years**	55 ± 14	56 ± 13	49 ± 15	***p*** < **0.001**
**Gender**				
Male	221 (43%)	183 (42%)	38 (47%)	0.44
Female	293 (57%)	250 (58%)	43 (53%)	
**Localization**				
Right colon	118 (23%)	81 (19%)	37 (46%)	***p*** < **0.001**
Left colon	220 (43%)	193 (44%)	27 (33%)	
Rectum	170 (33%)	155 (36%)	15 (19%)	
Right and left colon	6 (1%)	4 (1%)	2 (2%)	
**Time of tumor onset**				
Primary	430 (83%)	375 (87%)	55 (68%)	***p*** < **0.001**
Metachronous *	55 (11%)	32 (7%)	23 (28%)	
Synchronous **	29 (6%)	26 (6%)	3 (4%)	
**Stage**				
I	48 (10%)	39 (9%)	9 (11%)	***p*** < **0.001**
II	213 (41%)	169 (39%)	44 (54%)	
III	160 (31%)	134 (31%)	26 (32%)	
IV	93 (18%)	91 (21%)	2 (2%)	

*—A tumor that arose after 6 months and later from the primary (regardless of localization); **—a colon tumor diagnosed simultaneously with another tumor (regardless of localization) or no later than 6 months from detection.

**Table 2 ijms-23-07062-t002:** Clinical and morphological data of patients with Lynch syndrome and MSI in sporadic tumor.

	MSI *n* = 81	Lynch Syndrome *n* = 36	Non-Lynch Syndrome *n* = 45	*p*-Value
**Age, mean ± SD, years**	49 ± 15	43 ± 13	53 ± 14	***p* < 0.001**
**Gender**				
Male	38 (47%)	20 (56%)	27 (60%)	0.69
Female	43 (53%)	16 (44%)	18 (40%)	
**Localization**				
Right colon	37 (46%)	16 (44%)	21 (47%)	0.88
Left colon	27 (33%)	11 (31%)	16 (36%)	
Rectum	15 (19%)	8 (22%)	7 (15%)	
Right and left colon	2 (2%)	1 (3%)	1 (2%)	
**Time of tumor onset**				
Primary	55 (68%)	27 (75%)	28 (63%)	0.47
Metachronous *	23 (28%)	8 (22%)	15 (33%)	
Synchronous **	3 (4%)	1 (3%)	2 (4%)	
**Stage**				
I	9 (11%)	4 (11%)	5 (11%)	0.3
II	44 (54%)	23 (64%)	21 (47%)	
III	26 (32%)	9 (25%)	17 (38%	
IV	2 (2%)	0	2 (4%)	

*—A tumor that arose after 6 months and later from the primary (regardless of localization); **—a colon tumor diagnosed simultaneously with another tumor (regardless of localization), or no later than 6 months from detection.

**Table 3 ijms-23-07062-t003:** Univariate and multivariate analysis of risk factors for patients with MSI.

	Univariate			Multivariate		
Factor	OR	[95% CI]	*p*	OR	[95% CI]	*p*
**Age**	0.96	[0.94–0.98]	**<0.001**	0.91	[0.89–0.94]	**<0.001**
**Gender**						
Male	1			1		
Female	0.83	[0.51–1.33]	0.44			
**Localization**						
Rectum	1			1		
Right	4.72	[2.45–9.11]	**<0.001**	9.14	[4.0–20.87]	**<0.001**
Left	1.45	[0.74–2.81]	0.28			
Right and left colon	5.17	[0.87–30.58]	0.07			
**Time of tumor onset**						
Primary	1			1		
Metachronous *	4.9	[2.67–8.98]	**<0.001**	20.36	[8.15–50.87]	**<0.001**
Synchronous **	0.79	[0.23–2.69]	0.70			
**T (tumor)**						
1	1			1		
2	1.14	[0.26–4.89]	0.86			
3	1.10	[0.31–3.90]	0.88			
4	1.34	[0.37–4.81]	0.65			
**N (nodes)**						
0	1			1		
1	0.73	[0.40–1.34]	0.31			
2	0.36	[0.18–0.71]	**<0.001**	0.06	0.002–1.45	0.08
**Stage (I–IV)**						
I	1			1		
II	1.13	[0.51–2.50]	0.77			
III	0.84	[0.36–1.94]	0.68			
IV	0.10	[0.02–0.46]	**<0.001**	0.91	[0.07–11.54]	0.94

*—A tumor that arose after 6 months and later from the primary (regardless of localization); **—a colon tumor diagnosed simultaneously with another tumor (regardless of localization), or no later than 6 months from detection.

**Table 4 ijms-23-07062-t004:** Data patients of study.

	*n* = 514
**Age**, mean ± SD, years	55 ± 14
**Gender**	
Male	221 (43%)
Female	293 (57%)
**Localization**	
Right colon	118 (23%)
Left colon	220 (43%)
Rectum	170 (33%)
Right and left colon	6 (1%)
**Time of tumor onset**	
Primary	430 (84%)
Metachronous *	55 (11%)
Synchronous **	29 (5%)
**Stage**	
I	48 (9%)
II	213 (42%)
III	160 (31%)
IV	93 (18%)
**T (tumor)**	
T1	22 (4%)
T2	46 (9%)
T3	263 (51%)
T4	183 (36%)
**N (lymph node)**	
N0	273 (53%)
N1	105 (21%)
N2	136 (26%)
**Distant metastases**	
No	421 (82%)
Yes	93 (18%)

*—A tumor that arose after 6 months and later from the primary (regardless of localization); **—a colon tumor diagnosed simultaneously with another tumor (regardless of localization), or no later than 6 months from detection.

**Table 5 ijms-23-07062-t005:** Characteristics of markers used for fragment analysis.

Marker	Gene	Cytogenetic Location	Genomic Coordinates (GRCh38)	Primers 5′–3′	Fragment Length
NR21	*SLC7A8*	14q11.2	14:23,125,294–23,183,659	FAM–GAGTCGCTGGCACAGTTCTA R–CTGGTCACTCGCGTTTACAA	110
NR24	*ZNF2*	2q11.1	2:95,165,808–95,184,316	FAM–GCTGAATTTTACCTCCTGAC R–ATTGTGCCATTGCATTCCAA	129
BAT25	*KIT*	4q12	4:54,657,927–54,740,714	FAM–TCGCCTCCAAGAATGTAAGT R–TCTGCATTTTAACTATGGCTC	124
BAT26	*MSH2*	2p21–p16	2:47,403,066–47,634,500	FAM–TGACTACTTTTGACTTCAGCC R–AACCATTCAACATTTTTAACCC	122
NR27	*MAP4K3*	2p22.1	2:39,248,940–39,437,311	FAM–AACCATGCTTGCAAACCACT R–CGATAATACTAGCAATGACC	90

## Data Availability

The data presented in this study are available on request from the corresponding author.

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
