# Peer review of "Microsatellite Instability in Russian Patients with Colorectal Cancer"

_ijms, 2022, doi:10.3390/ijms23137062_

Round 1

Reviewer 1 Report

Manuscript No. ijms-1765194

„Microsatellite Instability in Russian Patients with Colorectal Cancer” for International Journal of Molecular Sciences

Comments:

1.      At the end of the Introduction section, please include a clearly defined purpose of your work.

2.      Table 1. Please correct the table caption. Patient’s characteristics according to MSI / MSS status. OR MSI vs MSS status. OR microsatellite stability (MS) status.

3.      What was the situation of the necessity to administer neoadjuvant or adjuvant therapy in the group of patients, depending on the MS status? Was there a clear correlation between the presence of metastases and the MS status and stage of disease? Was the patients' survival monitored depending on the MS status? Can the obtained results guide the choice of a therapeutic method?

4.      I am asking for the literature records unification.

Reviewer 2 Report

In the manuscript by Shubin et al., they analyzed the clinical data of 514 Russian patients to decide the frequency and characteristics of microsatellite instability in the Russian population. The results provided are of interest in general to clinicians and would be helpful to guide the diagnosis/categorizing/treatment of CRC patients. I have two suggestions for improving the manuscript:

1. The authors should provide the full name of "MMR" (supposedly mismatch repair) before using this acronym in the manuscript.

2. The authors should change Figures 4 and 6 to normal histograms. The way these two figures are plotted in the current manuscript is uncommon and distracting, and does not provide more information than regular histograms.

Round 2

Reviewer 1 Report

The manuscript is corrected.